# Efficient Knowledge Distillation via Salient Feature Masking

## Abstract

Traditional Knowledge Distillation (KD) transfers all outputs from a teacher model to a student model, often introducing knowledge redundancy. This redundancy dilutes critical information, leading to degraded student model performance. To address this, we propose Salient Feature Masking for Knowledge Distillation (SFKD), a lightweight enhancement that masks out less informative components and selectively distills only the top-K activations. SFKD is a drop-in modification applicable to both logit-based and feature-based KD, incurs negligible overhead, and sharpens the student's learning signal. Empirically, SFKD yields consistent gains across architectures (ConvNeXt, ViT) and scales (CIFAR-100: +5.44 pp; CUB: +6.39 pp; ImageNet-1K: +3.57 pp). We also provide intuition from the Information Bottleneck perspective to motivate why filtering out less salient teacher signals benefits the student. Overall, SFKD is a simple, empirically validated method for training student models that are both leaner and more accurate.

## 1 Introduction

While deep neural networks continue to grow in depth, width, and computational demands, the devices that ultimately rely on these algorithms – mobile phones, autonomous drones, and battery-constrained sensors – operate under tight budgets with respect to memory, energy, and latency. *Knowledge distillation (KD)* addresses this gap by transferring the behavior of a high-capacity *teacher* network to a compact *student*. Conventional pipelines, however, relay the full spectrum of teacher signals: the entire logit vector, intermediate feature and attention maps (Romero et al., 2014; Komodakis & Zagoruyko, 2017; Tian et al., 2019; Chen et al., 2021b). However, such indiscriminate transfer overwhelms the student model with peripheral or even misleading activations, thereby misguiding its limited capacity and hindering generalization (Ojha et al., 2023).

We reinterpret distillation through the lens of the *Information Bottleneck* (IB) principle (Saxe et al., 2018). Each teacher activation constitutes a noisy channel between the input–label pair $(X, Y)$ and a representation $F$. The IB objective seeks the most concise $F$ that maximizes $I(F; Y)$ while suppressing redundant information $I(X; F)$. From this perspective, only a subset of the teacher's knowledge is worth transmitting.

Guided by the IB principle, we derive **S**alient **F**eature masking for **K**nowledge **D**istillation (**SFKD**), a unified top-K masking rule that filters teacher signals before they reach the student. Viewing each teacher activation as a noisy communication channel, SFKD ranks logit entries, feature map channels, and attention coefficients by a lightweight mutual information proxy, and retains only the $K$ most informative elements. By discarding poor cues, the method suppresses transfer bias and compels the student to focus on the evidence most predictive of $Y$, thereby improving accuracy, robustness, and interpretability at negligible computational cost. Our contributions are as follows:

1. **Unified saliency mask for distillation.** We introduce SFKD, a single top-K masking rule that selects the most informative logits, feature-map values, and attention coefficients, and distills only these signals from teacher to student.

2. **Information-Bottleneck justification and guarantee.** By re-casting the mask selection as an Information-Bottleneck optimization, we prove that the retained activations maximize mutual

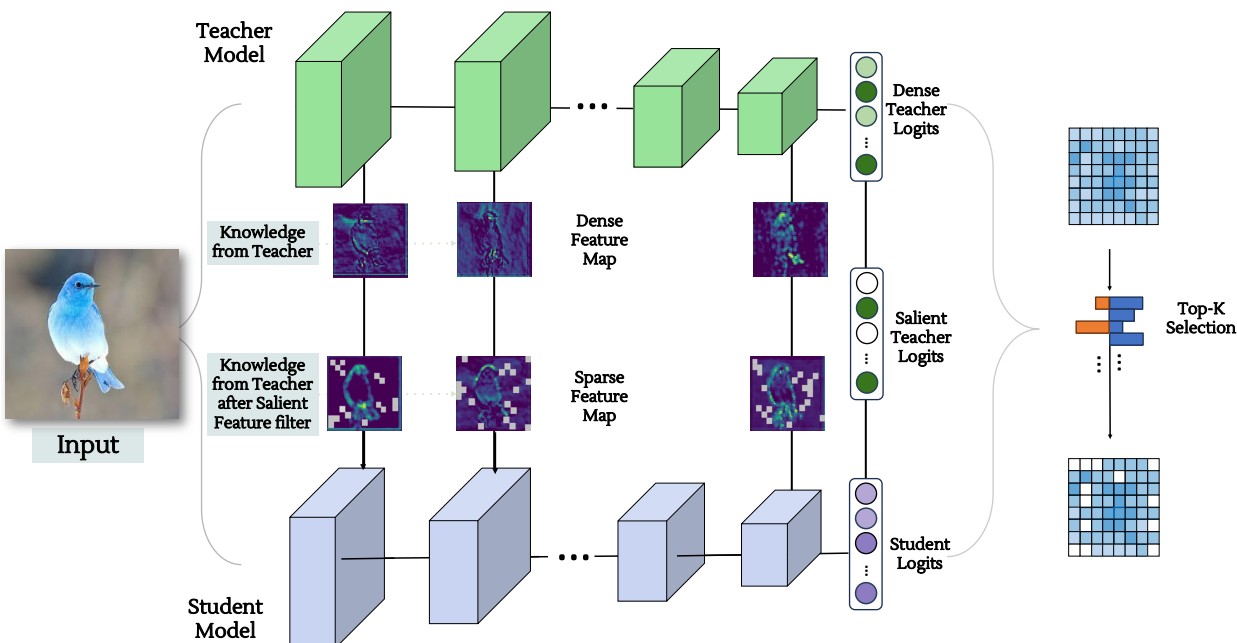

Figure 1: The concept of the proposed SFKD. SFKD distinctively concentrates on: 1) distilling critical classification knowledge, 2) transferring essential information from intermediate layers, and 3) refining attention mechanisms for knowledge distillation.

information $I(X; F_K)$ under a budget on student capacity, and we bound the information lost when discarding the remaining entries.

3. **SFKD drops straight into existing KD pipelines.** We find that SFKD consistently raises top-1 accuracy across CIFAR-100, CUB200, and ImageNet-1K setups, while adding negligible computational overhead.

## 2 Related Work

**Knowledge Distillation (KD)** variants generally fall into three categories based on the type of knowledge transferred: logits (Hinton et al., 2015; Furlanello et al., 2018; Mirzadeh et al., 2019; Zhao et al., 2022; Jin et al., 2023), features (Chen et al., 2021b; Heo et al., 2019; Park et al., 2019; Peng et al., 2019; Romero et al., 2014; Tung & Mori, 2019; Tian et al., 2019; Liu et al., 2023), and attention (Komodakis & Zagoruyko, 2017; Guo et al., 2023). Vanilla KD (Hinton et al., 2015) transfers class predictions from the teacher's output layer to guide the student's training. In contrast, feature distillation extracts knowledge from intermediate layers; for example, FitNet (Romero et al., 2014) aligns feature maps between specific teacher-student layers. Attention-based methods (Komodakis & Zagoruyko, 2017) use attention maps derived from feature representations for comprehensive knowledge transfer across layers. Subsequent studies explore applications of KD in semantic segmentation (Liu et al., 2019; Yang et al., 2022), object detection (Li et al., 2024; Zhang et al., 2024), and student architecture search (Dong et al., 2023).

**Information Bottleneck (IB)** is a principle introduced by (Tishby et al., 2000), which aims to extract the most relevant information from an input. The IB method defines a trade-off between compressing the input representation and preserving information about the target variable. The IB framework was extended to deep learning (Tishby & Zaslavsky, 2015), proposing that deep neural networks (DNNs) implicitly optimize this trade-off during training. (Shwartz-Ziv & Tishby, 2017) applied the IB principle to analyze the training dynamics of DNNs, showing that the learning process can be viewed as a progression from fitting the data to compressing irrelevant information, thereby enhancing generalization.

(Pogodin & Latham, 2020) further advanced this field by proposing learning rules based on the IB principle, achieving performance comparable to backpropagation in image classification tasks. More recently, (Wang et al., 2022) found that an intermediate model, often at an optimal training checkpoint, can serve as a more effective teacher than a fully converged model, despite its lower accuracy. In contrast, our work uniquely applies the IB principle to interpret the KD process. While (Goldfeld et al., 2019) analyzed mutual information compression in representation learning, we are the first to use the IB framework to specifically examine information flow during distillation, offering novel insights into the underlying dynamics of the process.

## 3 Methods

Let $\mathbf{a} \in \mathbb{R}^N$ denote a one–dimensional teacher activation (e.g., the class–logit vector, a flattened feature map, or a flattened attention tensor). Our goal is to retain only the $K$ most informative elements (as measured via mutual information) and suppress the rest. We define the top-$K$ index set as:

$$I_{Top-K} = \{\, i \in \{1, \ldots, N\} \mid \mathbf{a}_i \text{ is among the } K \text{ largest elements of } \mathbf{a} \,\} \tag{1}$$

From $I_{\text{Top-}K}$ we construct the binary mask $\mathbf{M} \in {0, 1}^N$ with components

$$M_i = \begin{cases} 1, & \text{if } i \in I_{\text{Top-K}} \\ 0, & \text{otherwise.} \end{cases} \tag{2}$$

Applying the mask to $\mathbf{a}$ via the element–wise (Hadamard) product $\odot$ yields the top-$K$ masked activation:

$$\mathbf{a}_K := \mathbf{M} \odot \mathbf{a}. \tag{3}$$

This operation leaves the $K$ salient entries untouched ($a_{K,i} = a_i$ for $i \in I_{\text{Top-}K}$) and zeros out all others ($a_{K,i} = 0$ otherwise). Whenever the activation vector is denoted $\mathbf{F}$ we write $\mathbf{F}_K = \mathbf{M} \odot \mathbf{F}$ for brevity.

Our findings have broad applicability, covering a wide range of distillation techniques, as illustrated in Figure 1. We focus on standard methods representing three main families of distillation approaches: *output-based* (Hinton et al., 2015), *feature-based* (Romero et al., 2014), and *attention-based* (Komodakis & Zagoruyko, 2017). The objectives of these methods are combined with the cross-entropy loss $L_{CLS}(z_s, y) := -\sum_{j=1}^{c} y_j \log \sigma_j(z_s)$, where $y$ is the ground-truth one-hot label vector, $z_s$ is the student's logit output, $\sigma_j(z) = e^{z_j} / \sum_i e^{z_i}$ is the softmax function, and $c$ is the number of classes.

**(1) Output-based:** The salient feature masking operates on the logit space by retaining only the top-K logits from the teacher's distribution based on their magnitude. The knowledge transfer is then performed through KL-divergence minimization between the masked teacher distribution and student predictions:

$$L_{KL}(z_s, z_{t^K}) := -\tau^2 \sum_{j=1}^{c} \sigma_j \left( \frac{z_{t^K}}{\tau} \right) \log \sigma_j \left( \frac{z_s}{\tau} \right), \tag{4}$$

where $z_{t^K}$ denotes the teacher's logits after top-K masking; $\tau$ is a scaling temperature; and the overall loss function is $\gamma L_{CLS} + \alpha L_{KL}$ with balancing parameters $\gamma$ and $\alpha$.

**(2) Feature-based:** The student's intermediate features $F_s^{(l)}$ are trained to mimic only the $K$ relevant features of the teacher's $F_{t^K}^{(l)}$ for a given image $X$ at layer $l$. The student's features are first projected via a transformation function $r$ to match the spatial dimensions or number of channels of the teacher's features (e.g., a linear projection layer to align the number of channels in $F_s$ with those in $F_t$). Their similarity is then optimized by minimizing the mean squared error:

$$L_{Hint}(F_s^{(l)}, F_{t^K}^{(l)}) = \frac{1}{2} ||F_{t^K}^{(l)} - r(F_s^{(l)})||_2^2. \tag{5}$$

The total loss is $\gamma L_{CLS} + \beta L_{Hint}$, where $\gamma$ and $\beta$ are balancing parameters. '*Hint*' represents all feature-based KD methods.

**(3) Attention-based:** Let $I$ be the set of indices representing the teacher-student activation layer pairs where attention maps are transferred. The total attention transfer loss is then defined as:

$$L_{AT} = L_{CLS} + \frac{\beta}{2} \sum_{j \in I} \left\| \frac{Q_s^j}{\|Q_s^j\|_2} - \frac{Q_{t^K}^j}{\|Q_{t^K}^j\|_2} \right\|_p \tag{6}$$

where $Q_s^j = \text{vec}(\phi(A_s^j))$ and $Q_{t^K}^j = \text{vec}(\phi(\text{Top}_K(A_t^j)))$ are respectively the $j$-th pair of student and top-$K$ elements in the teacher's attention maps in vectorized form. A mapping function $\phi$ maps a 3D activation tensor $A \in R^{C \times H \times W}$ to a spatial attention map. $\beta > 0$ is a balancing parameter and $\|x\|_p$ is the $\ell_p$ norm of vector $x$ (typically $\ell_2$ norm).

# 4 An Information-Theoretic Perspective On SFKD

In this section, we use the well-established Information Bottleneck (IB) theory as a conceptual lens to motivate and analyze SFKD. This perspective provides a clear intuition for why selectively distilling information, rather than transferring the teacher's entire knowledge base, can lead to more efficient and effective student models.

## 4.1 The IB Principle

Let $X$ and $Y$ denote the input and label random variables, and let $F$ be an intermediate representation generated by a parameterized encoder $p_\phi(F|X)$. The classical IB objective (Tishby & Zaslavsky, 2015; Shwartz-Ziv & Tishby, 2017) seeks a trade-off between compressing the input and preserving predictive information about the label:

$$\min_\phi \; I(X;F) - \zeta \, I(F;Y), \tag{7}$$

where $\zeta > 0$ controls the trade-off. We acknowledge that for deterministic networks, the mutual information $I(X;F)$ is technically infinite. Following common practice in IB analysis of deep neural networks, we use practical MI estimators that rely on well-established lower bounds, allowing us to qualitatively analyze the information flow during training (Shwartz-Ziv & Tishby, 2017; Ahn et al., 2019).

Viewing this through the IB lens, the goal of knowledge distillation should be to create a "bottleneck" that filters the teacher's knowledge before it is transferred to the student. This ensures the student focuses its limited capacity on the most salient information.

The IB principle inspires several hypotheses about how SFKD should affect the student's learning dynamics, which we can visualize on the "information plane" ($I(X;F)$ vs. $I(F;Y)$).

- **H1** *Less input compression* – By focusing only on salient features, SFKD allows the student to retain more information about the original input, leading to a higher $I(X;F_s)$.

- **H2** *Faster label informativeness* – By receiving a cleaner, more concentrated learning signal, the student model trained with SFKD should learn the relationship with the output labels more quickly, resulting in a steeper initial increase in $I(F_s;Y)$.

- **H3** *Intermediate K is best* – Transferring too little information (a very small K) or too much (no masking) is suboptimal. An ideal "elbow point" for K should exist, where the student's performance is maximized.

Our empirical results, visualized in Figure 2, strongly support these hypotheses. We observe that SFKD consistently guides the student to a better position on the information plane (H1) and accelerates the learning of label-relevant information (H2) compared to standard KD. Furthermore, our ablation on the value of $K$ confirms that an intermediate level of masking is indeed optimal (H3), validating the core idea of creating an information bottleneck.

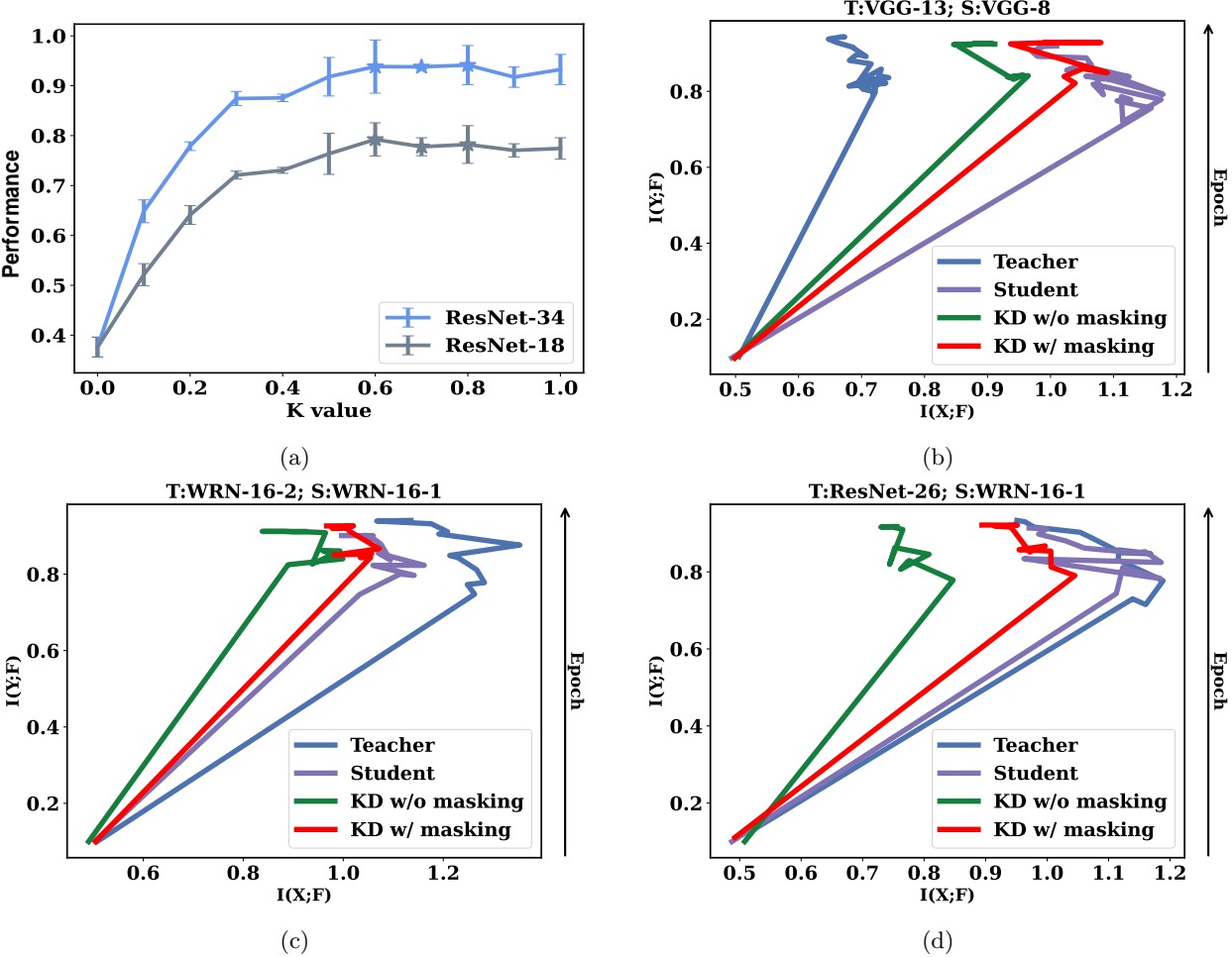

Figure 2: (a) An example of the optimal $K$ selection for salient feature masking. The horizontal axis shows the degree of Top-K sparsity ($K = 1$ meaning no masking). (b)–(d) The information plane for different teacher-student networks: the mutual information trajectories with respect to the training epochs.

# 5 Experiments

We demonstrate that our SFKD approach is method-agnostic by testing it across various existing distillation methods. Additionally, we show its two applications: (i) Selective Knowledge Sharing in Multi-Teacher Knowledge Distillation and (ii) Salient Feature Masking in Data-Free Knowledge Distillation. Furthermore, we perform ablation studies, implementing both as a standalone approach and in conjunction with the KD loss.

**Results on CIFAR-100.** Table 1 demonstrates the performance and robustness of SFKD when applied to similar vs. dissimilar model architectures. We extend our evaluation to more advanced network architectures, including ConvNeXt and Vision Transformers: *ViT-based teachers* distilled to both *CNN-based* and *ViT-based students*, with results presented in Table 2. This broader testing scope further validates the versatility of our method across diverse neural network designs to distill knowledge effectively across any architectures, such as from *ViTs* (Swin-T) to *CNN-based student* (ResNet-18) and from *ConvNeXt* to *Swin-P*[1].

**Results on ImageNet.** Table 3 reports Top-1 and Top-5 accuracies for SFKD ("Ours") compared to baseline methods (Chen et al., 2021b; Tian et al., 2019). Across teacher–student pairs of varying capacity

---

[1]Swin-Pico referred to as Swin-P

gaps, SFKD yields consistent gains in both metrics. These findings demonstrate the scalability of our approach to large-scale datasets, where maintaining signal quality during transfer is particularly challenging.

**Results on CUB200.** Table 4 evaluates SFKD on the fine-grained CUB200 bird classification task (Welinder et al., 2010), which requires discriminating between subtle inter-class variations. SFKD achieves substantial accuracy improvements across all tested configurations, indicating that its selective feature transfer enhances the capture of fine-grained discriminative cues.

Table 1: Comprehensive performance comparison on CIFAR-100. **Bold** indicates the best, underbar is the second-best value.

| Method | Same architecture style | | | | Different architecture style | | |
|---|---|---|---|---|---|---|---|
| T/S Pair | WRN-40-2 | WRN-40-2 | ResNet-32x4 | VGG-13 | VGG-13 | ResNet-32x4 | WRN-40-2 |
| | WRN-16-2 | WRN-40-1 | ResNet-8x4 | VGG-8 | MobileNetV2 | ShuffleNetV2 | ShuffleNetV1 |
| Teacher | 75.61 | 75.61 | 79.42 | 74.64 | 74.64 | 79.42 | 75.61 |
| Student | 73.26 | 71.98 | 73.09 | 70.36 | 64.60 | 71.82 | 70.50 |
| CAT-KD (Guo et al., 2023) (CVPR'23) | 75.60 | 74.82 | 76.91 | 74.65 | 69.13 | 78.41 | 77.35 |
| ReviewKD (Chen et al., 2021b) (CVPR'21) | 76.12 | 75.09 | 75.63 | 74.84 | 70.37 | 77.78 | 77.14 |
| DIST (Huang et al., 2022) (NeurIPS'22) | N/A | 74.73 | 76.31 | N/A | N/A | 77.35 | N/A |
| KD-Zero (Li et al., 2023a) (NeurIPS'23) | 76.42 | N/A | 77.85 | 75.26 | 70.42 | 77.45 | 77.52 |
| Auto-KD (Li et al., 2023b) (ICCV'23) | 76.86 | N/A | 77.61 | 75.36 | 70.58 | 77.52 | 77.46 |
| RLD (Sun et al., 2024b) (ICCV'25) | 76.02 | 74.88 | 76.64 | 74.93 | 69.97 | 77.56 | N/A |
| LS (MLKD+LS) (Sun et al., 2024a) (CVPR'24) | 76.95 | 75.56 | 78.28 | 75.22 | **70.94** | 78.76 | N/A |
| DKD (Zhao et al., 2022) (CVPR'22) | 76.24 | 74.81 | 76.32 | 74.68 | 69.71 | 77.07 | 76.70 |
| **DKD + SFKD** | 76.51 | 74.96 | 76.68 | 74.82 | 69.94 | 77.34 | 76.95 |
| SimKD (Chen et al., 2022) (CVPR'22) | 76.23 | 75.56 | 78.08 | 74.93 | 68.95 | 78.39 | N/A |
| **SimKD + SFKD** | 76.53 | **75.87** | **78.53** | 75.23 | 70.38 | 78.48 | **77.64** |
| MLKD (Jin et al., 2023) (CVPR'23) | 76.63 | 75.35 | 77.08 | 75.18 | 70.57 | 78.44 | 77.44 |
| **MLKD + SFKD** | **77.01** | 75.72 | 78.06 | **75.60** | 70.58 | **79.16** | 77.50 |

Table 2: SFKD with heterogeneous architectures on CIFAR-100: *ViT-based teachers* distilled to both *CNN-based* and *ViT-based students*.

| *ViT-based Teachers* | T. | Swin-T | ViT-S | Mixer-B/16 | ConvNeXt-T |
|---|---|---|---|---|---|
| | S. | ResNet-18 | ResNet-18 | ResNet-18 | Swin-P |
| | Teacher acc. | 89.26 | 92.43 | 87.62 | 88.41 |
| | Student acc. | 74.01 | 74.01 | 74.01 | 72.63 |
| Logit-based | DIST (Huang et al., 2022) | 77.75 | 76.49 | 76.36 | 76.41 |
| | KD (Hinton et al., 2015) | 78.74 | 77.26 | 77.79 | 76.44 |
| | **KD + SFKD** | **80.62**$_{+1.88}$ | **78.90**$_{+1.64}$ | **79.18**$_{+1.39}$ | **78.87**$_{+2.43}$ |

# 6 Discussion

Across CIFAR-100, ImageNet, and CUB200, SFKD consistently outperforms strong KD baselines, with relative gains up to +6.39 percentage points in accuracy on the CUB200 dataset. The breadth of tested teacher–student combinations—from homogeneous CNN–CNN settings to heterogeneous ViT–CNN and ConvNeXt–ViT transfers—demonstrates that SFKD's masking mechanism generalizes well to diverse architectural paradigms.

Furthermore, these performance improvements align with the Information Bottleneck perspective underpinning SFKD: by filtering out low-informative activations, the method reduces transfer noise and compels the

Table 3: Top-1 and Top-5 accuracy (%) on ImageNet validation.

| Teacher/Student | ResNet-34/ResNet-18 | | ResNet-50/MobileNet | |
|---|---|---|---|---|
| Accuracy | top-1 | top-5 | top-1 | top-5 |
| Teacher | 73.31 | 91.42 | 76.16 | 92.86 |
| Student | 69.75 | 89.07 | 68.87 | 88.76 |
| ReviewKD (Chen et al., 2021b) | 71.61 | 90.51 | 72.56 | 91.00 |
| SimKD (Chen et al., 2022) | 71.59 | 90.48 | 72.25 | 90.86 |
| CAT-KD (Guo et al., 2023) | 71.26 | 90.45 | 72.24 | 91.13 |
| AT (Komodakis & Zagoruyko, 2017) | 70.69 | **90.01** | 69.56 | 89.33 |
| **AT+SFKD** | $\textbf{70.84}_{+0.15}$ | 89.91 | $\textbf{70.88}_{+1.32}$ | $\textbf{90.00}_{+0.67}$ |
| KD (Hinton et al., 2015) | 70.66 | 89.88 | 68.58 | 88.98 |
| **KD+SFKD** | $\textbf{71.82}_{+1.16}$ | $\textbf{90.41}_{+0.53}$ | $\textbf{72.15}_{+3.57}$ | $\textbf{90.52}_{+1.54}$ |
| DKD (Zhao et al., 2022) | 71.70 | 90.41 | 72.05 | 91.05 |
| **DKD+SFKD** | $\textbf{72.10}_{+0.4}$ | $\textbf{90.70}_{+0.29}$ | $\textbf{72.95}_{+0.9}$ | $\textbf{91.30}_{+0.25}$ |

Table 4: Performance on the CUB200 dataset was evaluated across three teacher-student configurations: 1) identical structure but different sizes, 2) different architectures with equivalent depth, and 3) completely different networks in both architecture and depth.

| Teacher | ResNet-32x4 | ResNet-32x4 | VGG-13 | VGG-13 | ResNet-50 |
|---|---|---|---|---|---|
| Acc | 66.17 | 66.17 | 70.19 | 70.19 | 60.01 |
| Student | MobileNetV2 | ShuffleNetV1 | MobileNetV2 | VGG-8 | ShuffleNetV1 |
| Acc | 40.23 | 37.28 | 40.23 | 46.32 | 37.28 |
| SP (Tung & Mori, 2019) | 48.49 | 61.83 | 44.28 | 54.78 | 55.31 |
| CRD (Tian et al., 2019) | 57.45 | 62.28 | 56.45 | 66.10 | 57.45 |
| SemCKD (Chen et al., 2021a) | 56.89 | 63.78 | 68.23 | 66.54 | 57.20 |
| ReviewKD (Chen et al., 2021b) | - | 64.12 | 58.66 | 67.10 | - |
| KD (Hinton et al., 2015) | 56.09 | 61.68 | 53.98 | 64.18 | 57.21 |
| **KD+SFKD** | $\textbf{61.68}_{+5.59}$ | $\textbf{65.67}_{+3.99}$ | $\textbf{60.37}_{+6.39}$ | $\textbf{65.64}_{+1.46}$ | $\textbf{61.01}_{+3.8}$ |
| DKD (Zhao et al., 2022) | 59.94 | 64.51 | 58.45 | 67.20 | 59.21 |
| **DKD+SFKD** | $\textbf{62.15}_{+2.21}$ | $\textbf{67.09}_{+2.58}$ | $\textbf{61.49}_{+3.04}$ | $\textbf{68.88}_{+1.68}$ | $\textbf{63.99}_{+4.78}$ |

student to focus on the most predictive components. This not only yields higher accuracy but also enhances representation quality, as corroborated by t-SNE and class activation map analyses (Section 7.3). The results further suggest that SFKD is particularly beneficial in settings with limited student capacity or noisy supervision signals, such as data-free or multi-teacher distillation.

The empirical evidence provided supports SFKD as a lightweight, theoretically grounded enhancement to a wide range of KD frameworks, offering both practical performance gains and conceptual clarity on the role of selective knowledge transfer.

# 7 Advanced Application of SFKD

Beyond the standard single-teacher distillation framework, we demonstrate that SFKD's core principle provides significant advantages in more complex scenarios. In this section, we explore two such advanced applications: (1) enhancing knowledge transfer from an ensemble of models in Multi-Teacher Knowledge Distillation and (2) improving student performance in Data-Free Knowledge Distillation. Finally, we provide a series of visualizations that offer qualitative insights into *how* SFKD achieves its performance gains by improving the student's feature representations and focus.

Table 5: SFKD with Multi-Teacher Knowledge Distillation. The student models ShuffleNetV2 & VGG-8 were trained under the configuration of pre-trained Tri-ResNet-32x4.

| Teacher Networks | Student Network | S. | AEKD | SFKD + AEKD | SFKD + AEKD-F | Ensemble |
|---|---|---|---|---|---|---|
| Tri-ResNet-32x4 | ShuffleNetV2 | 71.82% | 75.87% | **76.17%** | **77.16**% | 81.31% |
| Tri-ResNet-32x4 | VGG-8 | 70.36% | 73.11% | **73.36%** | **73.80%** | 81.31% |

Table 6: Results of DFKD to various students on CIFAR-10.

| Teacher | Required data | VGG-11 | VGG-11 | ResNet-34 |
|---|---|---|---|---|
| Student | | VGG-11 | ResNet-18 | ResNet-18 |
| Student accuracy | Yes | 92.25% | 95.20% | 95.20% |
| Noise $\sim \mathcal{N}(0,1)$ | | 13.55% | 13.45% | 13.61% |
| DeepDream | No | 36.59% | 39.67% | 29.98% |
| DeepInversion (DI) | No | 84.16% | 83.82% | 91.43% |
| DI + SFKD ($K^{0.7}$) | No | **85.24%** | **84.86%** | **91.82%** |

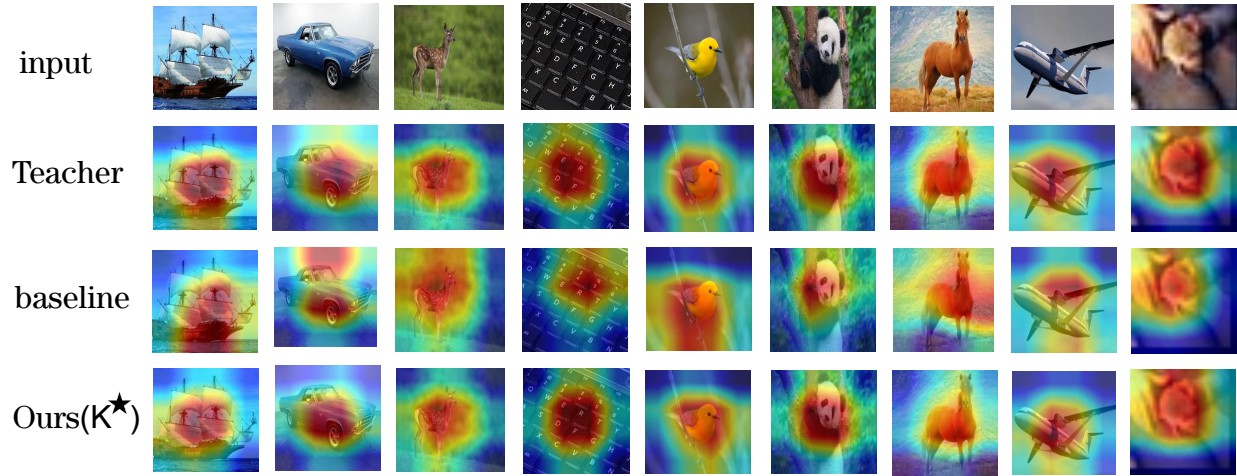

Figure 3: Class activation map of the distilled student model deployed with our method and baseline AT, the teacher model. The deeper the color, the more salient the corresponding feature of the image. The top row presents the input images, while the second, third, and fourth rows display the class activation maps of the teacher model, baseline AT ($K^1$), and SFKD ($K^\star$) respectively.

## 7.1 Application 1: Selective Knowledge Sharing in Multi-Teacher Knowledge Distillation

In multi-teacher distillation, conventional methods [(Du et al., 2020; You et al., 2017; Fukuda et al., 2017; Wu et al., 2019)] that average teacher outputs risk diluting specialized knowledge. SFKD avoids this pitfall by selectively distilling only the most salient signals from the teacher ensemble. This makes it uniquely suited for multi-teacher contexts, a claim supported by its superior accuracy in our experiments (Table 5). We demonstrate this by applying SFKD to the AEKD framework (Du et al., 2020), using a Tri-ResNet-32x4 ensemble to teach both VGG-8 and ShuffleNetV2 students. Across all configurations, SFKD consistently achieves the best performance by effectively channeling the most pertinent insights from the multiple experts.

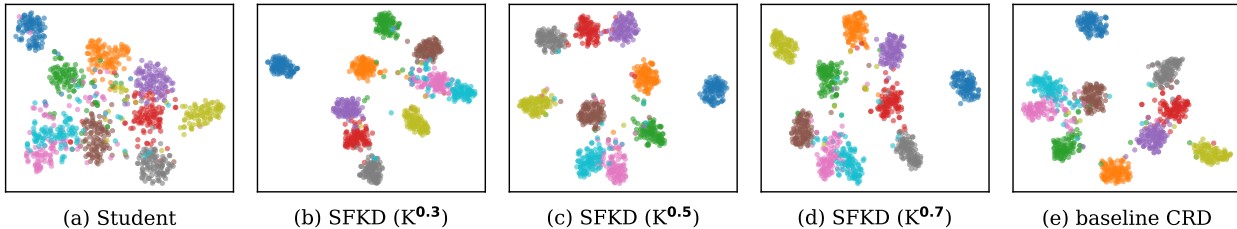

(a) Student     (b) SFKD ($K^{0.3}$)     (c) SFKD ($K^{0.5}$)     (d) SFKD ($K^{0.7}$)     (e) baseline CRD

Figure 4: t-SNE clustering: demonstrating model accuracy on CIFAR-100. 10 out of 100 classes were randomly sampled, as indicated by their respective colors. A high density of same-class dots and large separation among classes suggests better model classification accuracy.

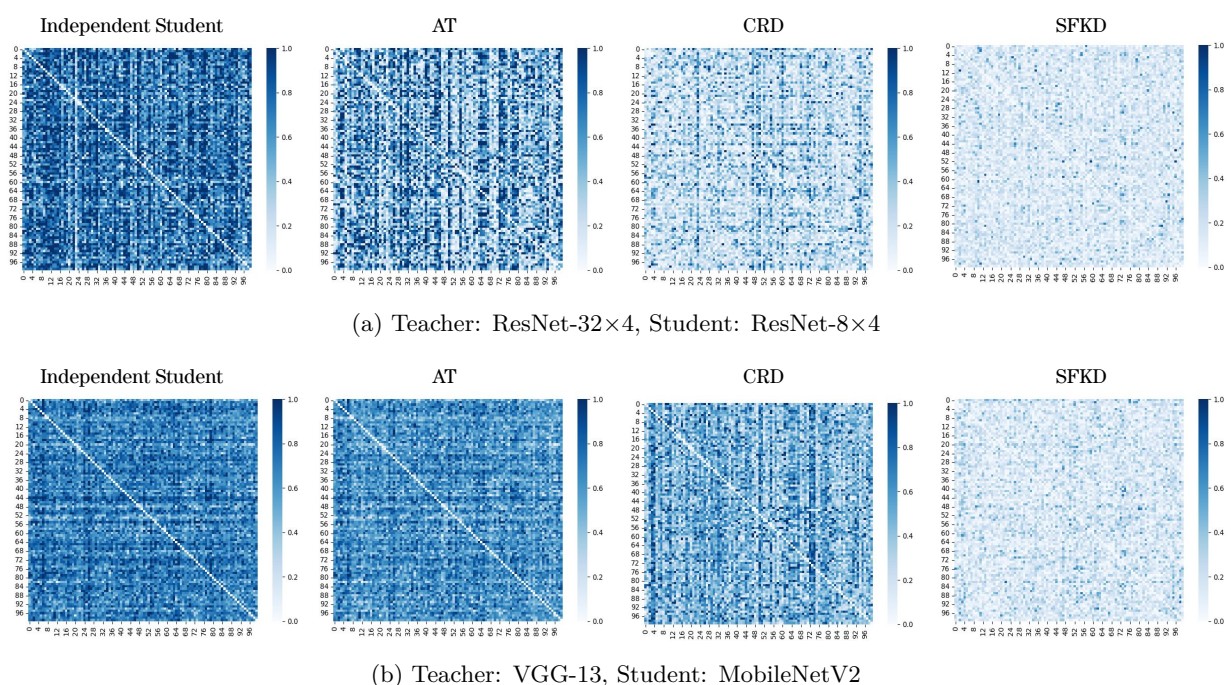

(a) Teacher: ResNet-32×4, Student: ResNet-8×4

(b) Teacher: VGG-13, Student: MobileNetV2

Figure 5: Contrast in correlation matrices of teacher and student classifier weights on CIFAR-100. The correlation matrices are computed using normalized weights.

## 7.2 Application 2: Salient Feature Masking in Data-Free Knowledge Distillation

Data-Free Knowledge Distillation (DFKD) relies on synthetic data, making it critical to filter out noise and artifacts. SFKD's methodology is particularly advantageous here, as it preserves the integrity of the distilled knowledge by focusing only on highly informative features. This targeted knowledge transfer helps the student model generalize better to real-world data. To validate this, we synthesized 100K images via DeepInversion (DI) (Yin et al., 2020) from CIFAR-10-trained VGG-11 and ResNet-34 teachers. As shown in Table 6, applying SFKD during distillation consistently improves the student's accuracy across all tested teacher-student pairs, highlighting its effectiveness in improving synthetic data utilization.

## 7.3 Visualization

To provide insight into *how* SFKD improves student models, we visualize and analyze the learned representations.

**Focused Attention with Class Activation Maps (CAMs).** We use CAMs (Zhou et al., 2016) to visualize where the model is "looking". Figure 3 contrasts the student model's attention when trained with a baseline method (AT) versus our AT+SFKD. The baseline model's focus often spreads to irrelevant background areas. In contrast, the SFKD-trained student concentrates its attention squarely on the target objects (car, bird, horse), closely mimicking the teacher's focus and demonstrating an improved ability to learn salient features.

**Improved Feature Separability with t-SNE.** To assess feature quality, we use t-SNE (van der Maaten & Hinton, 2008) to project the feature distributions of student networks trained on CIFAR-100 (ResNet-32x4 → ResNet-8x4). As shown in Figure 4, a student trained from scratch or with a baseline (CRD) exhibits significant class overlap. The student trained with SFKD, however, produces feature clusters that are far more compact and clearly separated, indicating a more discriminative and effective representation.

**Classifier Pattern Matching.** We further quantify the student's ability to learn the teacher's internal logic by measuring the L1 error between their classifier weight correlation matricesand illustrate this variance using a heatmap (Figure 5). Four methods were examined: the independent student without any distillation, alongside students trained with AT (Komodakis & Zagoruyko, 2017), CRD (Tian et al., 2019), and our approach, SFKD ($K^{0.3}$). The findings demonstrate that SFKD records the minimal difference across both sets of teacher-student pairs, showcasing SFKD's superior ability to replicate the teacher's correlation patterns.

## 8 Conclusion

In this work, we introduce salient feature masking for knowledge distillation, a simple but effective method that selectively distills the most pertinent features to enhance student performance. Compatible with existing KD variants, logit-based SFKD allows direct manipulation of a pre-trained network's logits by preserving high probability class values. This effective technique is easily applicable to large networks in real-world scenarios, which requires no retraining or modification of the original model. Leveraging the information bottleneck principle, we provide theoretical analysis and interoperability of SFKD's effectiveness, which explores insights into the teacher model's decision-making process. Our work opens up a few interesting research directions. First, it is intriguing to explore the characteristics of information flow during the distillation process. Second, finding the optimal $K$ value effectively without extensive tuning is important for the top-K salient feature distillation regarding heterogeneous teacher-student networks.

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

## Appendix

This supplementary document details mutual information estimation for $I(X;F)$ and $I(Y;F)$ (Section A), complete training setup with hyperparameters for CIFAR-100, CUB200, and ImageNet (Section B), and additional experiments including ablations, data-free knowledge distillation results, and representation visualizations (Section C).

## A   Mutual Information Estimation.

**Estimating $I(X;F)$.**  Let $R(X|F)$ denote the expected error for reconstructing $X$ from $F$. It is well known that $R(X|F)$ follows $I(X;F) = H(X) - H(X|F) \geq H(X) - R(X|F)$, where $H(X)$ is the Shannon entropy of $X$, which is a constant (Hjelm et al., 2019). Therefore, we estimate $I(X;F)$ by training a decoder parameterized by $w$ to obtain the minimal reconstruction loss, namely $I(X;F) \approx \max_w[H(X) - R_w(X|F)]$. In practice, we use the binary cross-entropy loss for $R_w(X|F)$.

**Estimating $I(Y;F)$.**  Since $I(Y;F) = H(Y) - H(Y|F) = H(Y) - \mathbb{E}_{(F,Y)}[-\log p(Y|F)]$, a straightforward approach is to train an auxiliary classifier $q_\psi(Y|F)$ with parameters $\psi$ to approximate $p(Y|F)$, such that we have $I(Y;F) \approx \max_\psi \{ H(Y) - \mathbb{E}_F[\sum_Y -p(Y|F)\log q_\psi(Y|F)] \}$. Finally, we estimate the expectation over $F$ using its sample mean $I(Y;F) \approx \max_\psi \{ H(Y) - \frac{1}{N}[\sum_{i=1}^N -\log q_\psi(Y_i|F_i)] \}$, where $\{(X_i, F_i, Y_i)\}_{i=1}^N$ are the samples. Consequently, $q_\psi(Y|F)$ can be trained in a regular classification fashion with the cross-entropy loss.

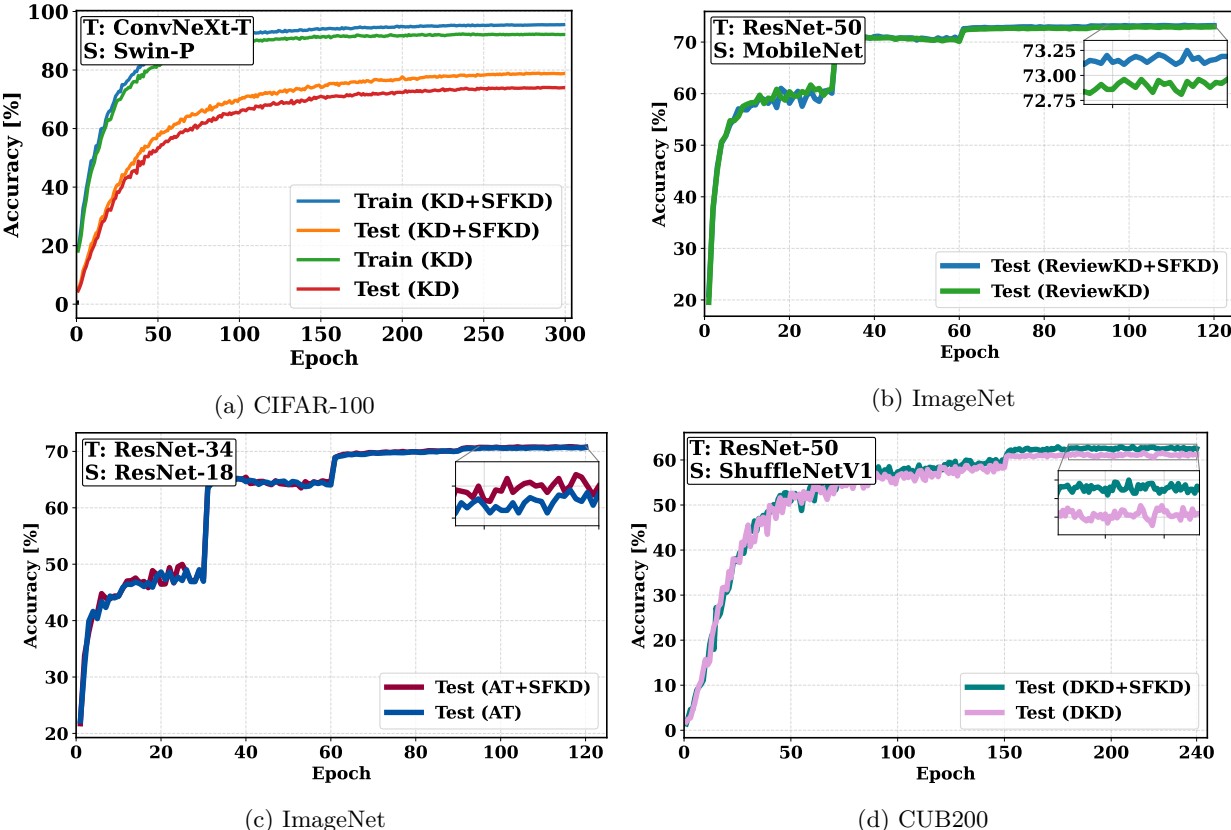

Figure 6: (a) ConvNeXt-T distilled to ViT-based student, evaluated on CIFAR-100. (b)–(c) ReviewKD and AT with our method, evaluated on ImageNet. (d) DKD with our approach, evaluated on CUB200.

## B   Experimental Settings

**Datasets and Baselines.** In this study, we utilize the CIFAR-100 (Krizhevsky & Hinton, 2009), CUB200 (Welinder et al., 2010) datasets and ImageNet (Deng et al., 2009). To demonstrate SFKD's versatility, we evaluate it with multiple KD methods: vanilla KD (Hinton et al., 2015), FitNet (Romero et al., 2014), AT (Komodakis & Zagoruyko, 2017), SP (Tung & Mori, 2019), CC (Peng et al., 2019), VID (Ahn et al., 2019), CRD (Tian et al., 2019), RKD (Park et al., 2019), PKT (Passalis & Tefas, 2018), DKD (Zhao et al., 2022) and Simple Knowledge Distillation (SimKD) (Chen et al., 2022). SFKD was implemented both as a standalone approach and in conjunction with the KD loss (except vanilla KD and SimKD) to demonstrate its efficacy and versatility. Experiments were performed using renowned backbone networks such as VGG (Simonyan & Zisserman, 2014), ResNet (He et al., 2016), Wide Residual Networks (WRN) (Zagoruyko & Komodakis, 2016), MobileNet (Sandler et al., 2018), ShuffleNet (Ma et al., 2018; Zhang et al., 2018) and more advanced networks, including ConvNeXt (Liu et al., 2022) and Vision Transformers (ViTs): ViT (Dosovitskiy, 2020) and Swin (Liu et al., 2021), across a range of teacher-student model pairings. To ensure a fair comparison with baseline methods, all training settings, including learning rate, batch size, and temperature, were standardized according to the baseline configurations.

**Training details:** We follow the conventional experimental settings of previous works (Tian et al., 2019; Zhao et al., 2022; Sun et al., 2024a;b) for CIFAR-100 and CUB200, training models for 240 epochs, except for MLKD being 480 as in (Jin et al., 2023; Sun et al., 2024a), with the learning rate being reduced by a factor of 10 at the 150th, 180th, and 210th epochs. The initial learning rate for architectures in the MobileNet/ShuffleNet series is 0.01, while it is 0.05 for all other architectures. A batch size 64 is used, alongside a weight decay of $5\times10^{-4}$ and stochastic gradient descent (SGD) optimizer. All results are presented as averages from 5 trials for homogeneous (Teacher/Student) T/S pairs and 3 trials for heterogeneous T/S pairs. We run ViT-based knowledge distillation processes for 300 epochs following the training scheme in (Hao et al., 2024). In IB analysis, we train the decoder to convergence with the Adam optimizer, with learning rate set to 0.05. All models on CIFAR-100 of the paper were run on NVIDIA GeForce GTX 1080 Ti GPUs (6 GPUs). Note: the default setting is for a single-GPU training. For ImageNet, the initial learning rate is set to 0.1 and then divided by 10 at 30th, 60th, 90th of the total 120 training epochs. We conducted experiments on ImageNet using 24 NVIDIA A100 GPUs.

## C   More Ablation Studies and Results

**More Experiments.**  Our approach effectively supports ViT distillation due to its model-agnostic nature, with experiments conducted on CIFAR-100 under the same conditions as (Hao et al., 2024). As shown in Table **2** of our main manuscript, our method consistently enhances KD performance across various *ViT-based, CNN-based*, and *MLP-based* (Mixer-B/16) models. Figure 6a illustrates both training and testing accuracy measurements comparing standard KD against KD enhanced with our SFKD method when distilling from ConvNeXt-T to *ViT-based student*. Additional validation on ImageNet (Deng et al., 2009) with ReviewKD (Chen et al., 2021b) was conducted on ResNet-50 to MobileNet, achieving Top-1 accuracy 73.25% with SFKD, as shown in Figure 6b. Validation on CUB200 dataset (Welinder et al., 2010), DKD with our approach in Figure 6d shows that it enhances the accuracy.

**Standard Deviation for CIFAR-100 Benchmark Results.** Sensitivity analyses involving a broader range of K values variability measured by standard deviation across multiple trials on the CIFAR-100 benchmark is provided in Table 7 for student and teacher models that share the same architecture, over five runs, and dissimilar architectural designs, over three runs.

**Ablation study.** To provide a deeper understanding of our approach, we conducted an ablation study exploring the impact of each technique both individually and in combination with KD. The results of this analysis can be found in Table 8. This analysis reveals how each technique affects the overall performance and how their interactions contribute to the final results.

Table 7: Comprehensive performance comparison on CIFAR-100. The table shows the classification accuracy (%) for various distillation methods and their performance when enhanced with SFKD. Teacher/student pairs are abbreviated for space (e.g., R32x4/R8x4 is ResNet-32x4/ResNet-8x4; MNV2 is MobileNetV2; SNV2 is ShuffleNetV2).

| T/S Pair | WRN-40-2/ WRN-40-1 | R32x4/ R8x4 | VGG13/ VGG8 | VGG13/ MNV2 | R32x4/ SNV2 |
|---|---|---|---|---|---|
| Teacher Acc. | 75.61 | 79.42 | 74.64 | 74.64 | 79.42 |
| Student Acc. | 71.98 | 73.09 | 70.36 | 64.60 | 71.82 |
| *Performance Format: Baseline Method / **Baseline + SFKD*** | | | | | |
| KD (Hinton et al., 2015) | 73.54 / $\mathbf{74.05}_{\pm0.22}$ | 73.33 / $\mathbf{74.41}_{\pm0.12}$ | 72.98 / $\mathbf{73.58}_{\pm0.23}$ | 67.37 / $\mathbf{68.30}_{\pm0.17}$ | 74.45 / $\mathbf{75.50}_{\pm0.08}$ |
| FitNet (Romero et al., 2014) | 72.24 / $\mathbf{72.60}_{\pm0.27}$ | 73.50 / $\mathbf{74.43}_{\pm0.26}$ | 71.02 / $\mathbf{72.35}_{\pm0.26}$ | 64.14 / $\mathbf{65.29}_{\pm0.13}$ | 73.54 / $\mathbf{75.30}_{\pm0.17}$ |
| AT (Komodakis & Zagoruyko, 2017) | 72.77 / $\mathbf{73.42}_{\pm0.18}$ | 73.44 / $\mathbf{73.71}_{\pm0.16}$ | 71.43 / $\mathbf{72.54}_{\pm0.32}$ | 59.40 / $\mathbf{60.82}_{\pm0.31}$ | 72.73 / $\mathbf{73.62}_{\pm0.27}$ |
| SP (Tung & Mori, 2019) | 72.43 / $\mathbf{73.51}_{\pm0.35}$ | 72.94 / $\mathbf{73.21}_{\pm0.07}$ | 72.68 / $\mathbf{73.23}_{\pm0.19}$ | 66.30 / $\mathbf{67.05}_{\pm0.29}$ | 74.56 / $\mathbf{76.20}_{\pm0.29}$ |
| CC (Peng et al., 2019) | 72.21 / $\mathbf{72.42}_{\pm0.15}$ | 72.97 / $\mathbf{73.17}_{\pm0.12}$ | 70.71 / $\mathbf{71.97}_{\pm0.30}$ | 64.86 / $\mathbf{65.58}_{\pm0.14}$ | 71.29 / $\mathbf{73.04}_{\pm0.36}$ |
| VID (Ahn et al., 2019) | 73.30 / $\mathbf{73.62}_{\pm0.18}$ | 73.09 / $\mathbf{73.39}_{\pm0.16}$ | 71.23 / $\mathbf{71.94}_{\pm0.22}$ | 65.56 / $\mathbf{65.72}_{\pm0.42}$ | 73.40 / $\mathbf{74.93}_{\pm0.07}$ |
| RKD (Park et al., 2019) | 72.22 / $\mathbf{72.56}_{\pm0.24}$ | 71.90 / $\mathbf{72.59}_{\pm0.28}$ | 71.48 / $\mathbf{71.66}_{\pm0.23}$ | 64.52 / $\mathbf{65.58}_{\pm0.21}$ | 73.21 / $\mathbf{74.13}_{\pm0.38}$ |
| PKT (Passalis & Tefas, 2018) | 73.45 / $\mathbf{73.83}_{\pm0.20}$ | 73.64 / $\mathbf{74.36}_{\pm0.17}$ | 72.88 / $\mathbf{73.19}_{\pm0.21}$ | 67.13 / $\mathbf{68.03}_{\pm0.20}$ | 74.69 / $\mathbf{75.84}_{\pm0.35}$ |
| CRD (Tian et al., 2019) | 74.14 / $\mathbf{74.43}_{\pm0.29}$ | 75.51 / $\mathbf{75.80}_{\pm0.19}$ | 73.94 / $\mathbf{74.08}_{\pm0.07}$ | 69.73 / $\mathbf{69.84}_{\pm0.27}$ | 75.65 / $\mathbf{76.33}_{\pm0.26}$ |
| SimKD (Chen et al., 2022) | 75.56 / $\mathbf{75.87}_{\pm0.21}$ | 78.08 / $\mathbf{78.53}_{\pm0.24}$ | 74.93 / $\mathbf{75.23}_{\pm0.08}$ | 68.95 / $\mathbf{70.38}_{\pm0.31}$ | 78.39 / $\mathbf{78.48}_{\pm0.13}$ |

Table 8: Individual and joint contributions to performance are illustrated through feature-based and logit-based combinations. The baseline method represents a feature-based approach, while KD indicates a logit-based method.

| Method/T-S pair | WRN-40-2/ShuffleNetV1 | | | | | |
|---|---|---|---|---|---|---|
| SP (Tung & Mori, 2019) | ✓ | | | | | |
| SP + SFKD | | | ✓ | | | |
| SP + KD | | ✓ | | | | |
| SP + (KD+SFKD) | | | | | ✓ | |
| (SP + SFKD) +KD | | | | ✓ | | |
| (SP + SFKD) + (KD+SFKD) | | | | | | ✓ |
| | 74.52 | 75.56 | 76.11 | 76.76 | 76.63 | 76.68 |

## C.1 Data-Free Knowledge Distillation.

We extend our investigation to the domain of Data-Free Knowledge Distillation (DFKD) specifically to evaluate our method's robustness when dealing with potentially degraded and/or suboptimal feature maps. While our method is designed to leverage high-quality feature maps from well-trained teacher models, we recognize that such optimal conditions may not always be available in real-world applications. Through DFKD experiments, we deliberately test our approach in scenarios where feature map quality is inherently compromised due to the synthetic nature of the training data.

Using DeepInversion (DI) (Yin et al., 2020), we synthesize 100K CIFAR-10 images from teacher models VGG-11 and ResNet-34. To comprehensively assess how our method performs with these potentially degraded feature maps, we employ multiple evaluation metrics: (a) single-value measures including Inception Score (IS) (Salimans et al., 2016) and Frechet Inception Distance (FID) (Heusel et al., 2017), and (b) two-value measures such as Precision and Recall (P&R) (Sajjadi et al., 2018). These metrics help quantify both the quality degradation in synthetic data and our method's resilience to such degradation. Table 9 presents a comparative analysis between our synthesized images and those generated by WGAN-GP, a baseline GAN-based model trained on original data.

Table 9: Metric result of synthesized images. A higher score of IS, Precision and Recall is better, whereas a lower score of FID is better.

| CIFAR-10 | | | | |
|---|---|---|---|---|
| **Inverted Model** | **IS ↑** | **FID ↓** | **Precision ↑** | **Recall ↑** |
| VGG-11 | 2.91 | 176.76 | 0.3824 | 0.0022 |
| ResNet-34 | 4.21 | 99.79 | 0.5824 | 0.1928 |
| WGAN-GP (Gulrajani et al., 2017) | 7.86 | 29.30 | 0.7040 | 0.4353 |

## C.2 More Visualizations

In Figure 7, we present visualizations comparing feature representations from models trained with our proposed distillation method (SFKD), alongside those from a teacher model, a student model trained without distillation, and CRD (Tian et al., 2019). The visual evidence in Figure 7 demonstrates that combining CRD with SFKD results in more distinct and separable features compared to the original representations, suggesting that SFKD enhances the distinguishability of deep features within the student model.

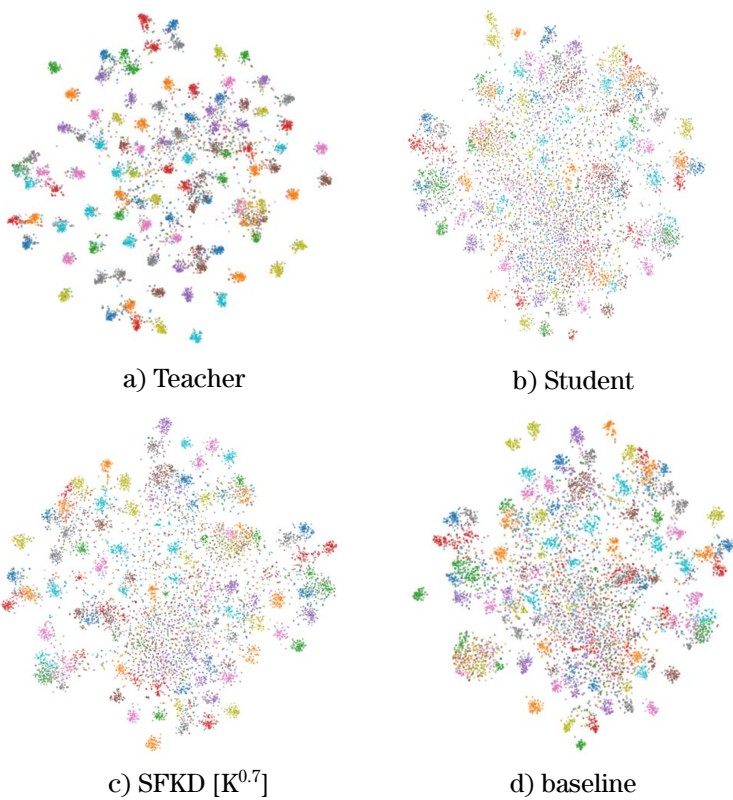

a) Teacher

b) Student

c) SFKD [$K^{0.7}$]

d) baseline

Figure 7: t-SNE clustering: demonstrating model accuracy on CIFAR-100. Points with the same color indicate they are from the same category, highlighting the model's proficiency in distinguishing between classes. A model that groups data points closely within the same class while keeping them widely separated from points of other classes demonstrates effective classification performance.

