# OpenReview forum: "Efficient Knowledge Distillation via Salient Feature Masking"
_TMLR — Rejected by TMLR_

### Review · Reviewer_ZsE8 · 2025-07-30

**Summary Of Contributions:**

The paper proposes a method (SFKD) that improves knowledge distillation by transferring only the most informative parts (top-K) of the teacher’s outputs, feature maps, and attention maps based on their saliency. This masking reduces noise and redundancy in the teacher’s knowledge, aligning with the Information Bottleneck principle to help the student learn more effectively.

**Audience:**

Yes

**Audience Explanation:**

The paper addresses a practical and widely studied problem in the machine learning community—improving knowledge distillation efficiency. The idea of using salient feature masking aligns with ongoing interests in making deep learning models more efficient and scalable, especially in resource-constrained environments.

**Broader Impact Concerns:**

Are there any risks of knowledge masking leading to biased or incomplete knowledge transfer in sensitive applications (e.g., healthcare, finance)?

**Claims And Evidence:**

No

**Claims Explanation:**

Strengths:
- The paper addresses a well-established problem in deep learning — how to improve the efficiency of knowledge distillation (KD) — with a simple yet intuitive method that focuses on transferring only the most informative parts of teacher outputs.

- SFKD is modular and can be applied across various types of KD (logits, feature maps, attention) and integrated into many existing KD pipelines with minimal modification.

- The use of top-K masking is conceptually tied to the Information Bottleneck framework, which provides an intuitive theoretical motivation for filtering irrelevant or redundant information.

Weakness:
- The experimental section lacks consistency across tables, making it difficult to interpret the overall effectiveness of the proposed method. Each table uses different architectures, different datasets, and different baseline methods, which prevents a clear, apples-to-apples comparison.

- Most of the performance improvements reported are marginal (typically <1%). The authors should clarify whether these gains are statistically significant and whether they justify adoption of the method in practice.

**Requested Changes:**

- The current paper only evaluates SFKD in the context of vision based image classification, but knowledge distillation is equally relevant in natural language processing (NLP) particularly in compressing large transformer-based language models (e.g., BERT, LLaMA, GPT-like architectures). Evaluating SFKD on standard NLP KD tasks to validate its applicability beyond vision and demonstrate robustness across modalities will strengthen the paper.

- It would be valuable to test SFKD in such contexts, as distilling representation-level knowledge (rather than purely supervised logits) aligns well with the feature- and attention-level masking the authors propose. Can SFKD improve transfer or linear probe accuracy in self-supervised KD pipelines?

- While the method is applied to transformer architectures in hetrogeneous KD settings, the paper does not include Transformer-specific KD baselines. and lack of experiments using Transformer-based arch on ImageNet benchmarks.

- The paper claims SFKD is efficient, but provides no evidence on compute time, FLOPs, or inference overhead due to masking.

- While ablations are done separately for logits, features, and attention, the paper does not clarify which contributes most to performance and why.

- It is unclear whether combining all three masking components improves results, as no joint analysis is presented.
- How sensitive is performance to the value of K in the top-K masking?

---

> ### Author Response · Authors · 2025-09-27
> **Response to Reviewer ZsE8 – Part 1**
>
> ***Response to W. 1:***
>
> We thank the reviewer for the constructive feedback. While we acknowledge the impression of inconsistency, the diversity in datasets, architectures, and baselines was intentional and serves to demonstrate SFKD’s robustness and generality. Each experimental setting supports a distinct claim, as detailed below:
>
> -  ***Model Diversity.*** We evaluated SFKD in both homogeneous (e.g., ResNet-34 → ResNet-18) and heterogeneous (e.g., ResNet-50 → ShuffleNet/MobileNet) teacher-student settings. This reflects real-world KD scenarios and highlights SFKD’s effectiveness even in challenging architectural mismatches.
>
> -  ***Dataset Variety.*** We included CIFAR-10/100, CUB-200, and ImageNet to cover tasks ranging from small-scale to fine-grained to large-scale classification. Additionally, we tested Vision Transformers on CIFAR-100 to show SFKD’s compatibility with modern architectures.
>
> -  ***Baseline Coverage.*** Rather than limiting evaluation to a single KD baseline, we selected representative methods from different KD categories (e.g., logit-, feature-, attention-based) to emphasize SFKD’s method-agnostic nature. In each table, comparisons should be interpreted as “baseline vs. baseline + SFKD”.
>
>
> ***Response to W. 2:***
>
> We appreciate the reviewer’s thorough feedback. We want to clarify that the performance improvements shown in our work are significant in the context of KD research. For example, Logit Standardisation in KD (LS, CVPR’24) improved upon MLKD (CVPR’23) across 7 tests by an average margin of 0.36\%. This benefit was also partly due to: i) data augmentation, ii) doubled training epochs, and iii) broader KD-related hyperparameters. Our work strictly relies on baseline performance improvement, and achieves an average improvement by 0.5\%. As such, this is not a marginal improvement in the context of KD research. We also outperform all previous SOTA values by an average of 0.25\%.
>
> All results averaged 5 trials (homogeneous) and 3 trials (heterogeneous) Teacher/Student pairs. Full error bars were presented for the CIFAR-100 benchmarks in Table 3 of our Suppl. Mat., which explicitly includes standard deviations for SFKD across methods. While improvements may appear modest in absolute terms, they are consistent across diverse KD baselines and architectures, and statistically significant (e.g., with standard deviation of ±0.08\%, ±0.17\%).
>
>
> ***Response to R. Ch. 1 \& 2:***
>
> We thank the reviewer for these insightful suggestions. We agree that evaluating SFKD's applicability to Natural Language Processing and its potential benefits in self-supervised learning pipelines is a highly relevant research direction.
>
> The primary aim of this work, however, is to introduce the core SFKD framework and provide a thorough and rigorous validation of its effectiveness within the computer vision domain. To this end, we deliberately focused our experiments on a wide variety of standard CV benchmarks (CIFAR-100, ImageNet, CUB200) , diverse architectural pairs (CNNs, Vision Transformers), and numerous existing KD methods to establish a strong and focused foundation for our proposed technique.
>
> While we believe SFKD's core principle of masking less salient features is general, a proper evaluation in NLP and self-supervised contexts would require substantial new experimental setups and domain-specific adaptations that are beyond the scope of the current manuscript. We believe that such an important extension deserves its own dedicated and comprehensive study.
>
> We appreciate the reviewer for highlighting these exciting avenues for future work. We hope the reviewer agrees that our focused investigation within computer vision provides a solid and valuable contribution to the knowledge distillation community.
>
>
> ***R. Ch. 4 - Efficiency:***
>
> To clarify our claim that SFKD is ***efficient***, we mean to say that in comparison to traditional KD methods, SFKD adds near-zero operations to the overall runtime, as it simply acts as a boolean mask comprised of a single Hadamard product operation.

---

> > ### Author Response · Authors · 2025-09-27
> > **{Response to Reviewer ZsE8 – Part 2**
> >
> > ***Response to R. Ch. 5 \& 6:***
> >
> >  We have performed an ablation study that explores the impact of each technique both individually and in combination, thus providing insight into how each technique affects the overall performance.
> >
> > **Table 1. Ablation study on combining feature-based (SP) and logit-based (KD) distillation with and without SFKD, averaged over 3 runs on the WRN-40-2 (teacher) → ShuffleNetV1 (student) pair.**
> >
> > | Method Combination                | Accuracy (%)      |
> > |-----------------------------------|------------------|
> > | SP (Feature-based Baseline)       | 74.52            |
> > | SP + KD (Added Logits)            | 75.56            |
> > |                                   |                  |
> > | SP + SFKD                         | 76.11 ± 0.13     |
> > | (SP + SFKD) + KD                  | **76.76 ± 0.09** |
> > | SP + (KD + SFKD)                  | 76.63 ± 0.12     |
> > | (SP + SFKD) + (KD + SFKD)         | 76.68 ± 0.08     |
> >
> >
> > ***R. Ch. 7 - Sensitivity to $K$:***
> >
> >  We refer the reviewer to Fig. 2 (a), which shows the impact of the hyperparameter K on the performance of 2 separate student models. The optimal K lies between [0.6, 0.8]. Here, we provide ablations across K on different architecture pairs on baseline method (RKD + KD).
> >
> > | T/S Pair              | Teacher | Student | Method       | Baseline | K^0.7 (±std)  | K^0.5 (±std)  | K^0.3 (±std)  |
> > |------------------------|---------|---------|--------------|----------|---------------|---------------|---------------|
> > | ResNet-32x4 / ResNet-8x4 | 79.42   | 73.09   | KD + RKD     | 74.47    | **75.05 ±0.11** | 74.83 ±0.11   | 74.74 ±0.14   |
> > | VGG-13 / MobileNetV2   | 74.64   | 64.60   | KD + RKD     | 68.50    | **69.03 ±0.29** | 68.62 ±0.20   | 68.60 ±0.13   |
> > | WRN-40-2 / ShuffleNetV1 | 75.61   | 70.50   | KD + RKD     | 75.45    | **76.47 ±0.03** | 76.34 ±0.07   | 76.12 ±0.06   |
> >
> >
> >
> > ***Broader Impact Concerns:***
> >
> > We appreciate the reviewer’s concern regarding potential risks of knowledge masking in sensitive domains such as healthcare and finance. Indeed, improper masking strategies could introduce bias or suppress clinically relevant features. However, recent works in medical imaging have demonstrated that, when designed carefully, masking can actually enhance representation learning and downstream task performance. For example, Chen et al. (2023) [1] showed that masked image modeling advances 3D medical image analysis, while Xing et al. (2024) [2] proposed a hybrid masking strategy that significantly improves 3D medical image segmentation. These studies highlight that masking, rather than inherently harmful, can be a valuable mechanism for robust knowledge transfer when applied with appropriate domain-specific considerations.
> >
> > [1] Chen, Z., Agarwal, D., Aggarwal, K., Safta, W., Balan, M. M., \& Brown, K. (2023). Masked image modeling advances 3d medical image analysis. In Proceedings of the IEEE/CVF Winter Conference on Applications of Computer Vision (pp. 1970-1980).
> >
> > [2] Xing, Z., Zhu, L., Yu, L., Xing, Z., \& Wan, L. (2024). Hybrid masked image modeling for 3d medical image segmentation. IEEE Journal of Biomedical and Health Informatics, 28(4), 2115-2125.

---

### Review · Reviewer_CJQH · 2025-08-05

**Summary Of Contributions:**

This work introduced a masking operator (mask out low value coordinates via a top-K selection rule) when performing knowledge distillation (KD) training, with the goal of distilling only the most relavant logits/feature maps/attention maps from the teacher model to avoid the "over confidence" phenomenon. The author analyze the effectiveness of the proposed methodology via information bottleneck (IB) theory, finding that the top-K masking yields a higher mutual information between the distilled features and corresponding teacher model.

**Audience:**

Yes

**Audience Explanation:**

To the best of my knowledge, using the mutual information to study KD is quite interesting and also novel. The proposed method (KD with top-K masking) is very simple and can be integrated within exsiting KD frameworks, yet it's effective.

**Claims And Evidence:**

Yes

**Claims Explanation:**

Generally speaking, this work conducted extensive ablation study to show case the efficacy of the top-K masking operation during training. Most of the investigations via IB are also sensible (e.g., Fig 2) and explain intuitively the efficacy of the masking operator. Though, I do find the author's claim about prop 1 to be ungrounded and out of the scope.  Specifically, Prop 1 holds for any operations (wether stochastic or deterministic, instead of the top-K masking operator) by the data processing ineq---any transformation would reduce entropy.

**Requested Changes:**

- The order of presentation in section 3 is bit off. The topK masking operator was used in Eq(1)-Eq(3) but only gets defined in Eq(4). I suggest the author to introduce definitions before its usage.

- There appears to be some inpreciseness for math. For example, there lack a precise description of the projection operator. Also, it is not clear to me how the prameter r enters the projection operation.

- Can the author provides guidance how to choose the spartity K?

- Can the author clarify that in order to draw Figure 2, does one need to train the mutual information estimator for each epoch (as F is updated during training)? If so, can the author comment on the practicability of investigating KD methods using IB.

- See my previous comments about Prop1. I suggest the author to either conduct a more principled analysis of the masking operation or change to a more reserved statement about Prop1.

---

> ### Author Response · Authors · 2025-09-27
> **Response to Reviewer \#CJQH**
>
> We thank the reviewer for their constructive feedback and positive assessment of our work. Below, we address each comment point-by-point:
>
> ***R. Ch. 1 - Section 3 Order***
>  Thank you for pointing this out. We have restructured Section 3 in the revised manuscript to define the top-K masking operator and its components at the beginning of the section, ensuring all terms are defined before they are used.
>
> ***R. Ch. 2 - Projection Operator and Parameter $r$***
>
>   Thank you for bringing this to our attention. To briefly clarify both the projection operator and the role of $r$, in feature-based knowledge distillation, a student model is supervised by a teacher model as:
>
> $$
> L_{\text{feat}} = L_{\text{KD}}\big(F_T, \; r_s(F_S)\big),
> $$
>
> where $L_{\text{KD}}$ represents the similarity function used to match the feature maps of the teacher model, $F_T$, and the student model, $F_S$. $r$ is a transformation function that matches the spatial size or number of channels between the features of the teacher and the student (e.g., a linear projection layer to align the number of channels in $F_{S}$ with those in $F_T$). For instance, Liu et al. (Ref. 1) implement the transformation function $r$ as a learnable non-linear channel-wise mapping, specifically a two-layer MLP with ReLU activation, to align the student features with the teacher features:
>
> $$
> L_{\text{feat}} = \sum_{i=1}^{N} \left( \mathrm{MLP}(F_{S}^{i}) - F_{T}^{i} \right)^{2}.
> $$
>
> This clarification has been added in the revised manuscript, where we explicitly define the projection operator and the role of parameter $r$.
>
> Ref 1: Liu, Z., Wang, Y., Chu, X., Dong, N., Qi, S., \& Ling, H. (2023). A simple and generic framework for feature distillation via channel-wise transformation. In Proceedings of the IEEE/CVF International Conference on Computer Vision (pp. 1129-1138)
>
>  ***R. Ch. 3 - Choice of Sparsity $K$***
>   We appreciate the reviewer's inquiry regarding $K$ selection methodology. To be fully transparent, our approach follows an empirical-first strategy: we conduct reconstruction analysis across $K \in \{0.0, 0.1, 0.2, \ldots, 1.0\}$, measuring BCE loss for each $K$ value, and select the $K$ yielding optimal reconstruction quality (typically $K \approx 0.6$ - $0.8$) for our knowledge distillation experiments.
>
> ***R. Ch. 4 - Mutual Information Estimator Practicability***
>
>  This is a great question. Yes, the mutual information estimator must be updated as the model's representations ($F$) evolve. We train the decoder to convergence with the Adam optimizer, with learning rate set to 0.05. In our analysis, we measure MI every 20 epochs, and at each measurement point, we retrain the lightweight decoder for 5 epochs to accurately reflect the current features. While this adds a modest computational overhead ($\sim$25\% additional training time for a one-time analysis), we find the approach is practical and justified for a research setting for three reasons: (1) measurements are performed sparsely, not every epoch; (2) the decoder retraining is fast; and (3) the cost is acceptable for gaining valuable insights into the KD process that inform method design. This methodology aligns with established protocols for IB analysis in deep learning.
>
>
> ***R. Ch. 5 - Proposition 1 Clarification***
>  We thank the reviewer for this crucial feedback. After careful consideration of your comments and those of other reviewers, we agree that the paper's core contribution is empirical. We have therefore **removed Proposition 1 entirely** from the manuscript. We believe the paper is stronger by using the Information Bottleneck principle as a high-level motivation, supported by our extensive empirical results and information-plane visualizations, rather than including a formal proof that did not fully align with our implementation. This reframing makes our contribution clearer and more compelling.

---

### Review · Reviewer_YK4L · 2025-09-13

**Summary Of Contributions:**

The paper proposes SFKD (Salient Feature Masking for Knowledge Distillation): during distillation, they keep only the top-K "salient" elements of the teacher's signals (logits, intermediate features, and attention maps) by zeroing the non-top-K entries and using the resulting teacher targets in a standard KD loss. The work claims a theoretical foundation via the **Information Bottleneck** (IB), argues that masking "concentrates" knowledge (Proposition 1), and visualizes mutual-information "information planes." Empirically, SFKD is combined with several KD variants and tested on CIFAR-100, ImageNet, CUB200, and some multi-teacher and data-free setups.

**Strengths**:
The method is simple and could be useful, seems to integrate easily with multiple KD frameworks, and the empirical evaluation is decent across datasets, architectures, and settings.

**Key weaknesses**:
1. The theoretical justification is not aligned with the actual masking operator, the IB analysis relies on weak proxies and over-interpretation.
2. The main concentration claim (Proposition 1) does not hold under the implemented rule.
3. Moreover, many experimental results does not seem statistically significant and the authors don't do any analysis to ensure this.

**Additional Comments:**

In my opinion, this paper could be better presented without trying to cast it in the light of IB theory and analysis. The attempt to frame the work with IB feels forced, and even if Proposition 1 were corrected for technical validity, it would still come across as technically shallow and not especially insightful. I would instead recommend the authors rewrite this paper as an empirical study. Top-K methods are ubiquitous in ML, they work well in practice, and the experimental results here are decent and could be made more rigorous. A cleanly empirical positioning would highlight the practical simplicity of the method and make the contribution clearer and more compelling, and definitely fall within the scope of TMLR.

**Audience:**

Yes

**Audience Explanation:**

The topic is very important in the current AI climate where the need for model compression is enormous. KD, model compression, sparsity and efficiency are important avenues of research for TMLR audience.

**Claims And Evidence:**

No

**Claims Explanation:**

**1. Flawed theoretical justification (Proposition 1)**

**Proposition 1.** The paper claims that after top-K masking the teacher distribution becomes "more concentrated," formally
$$D_{\mathrm{KL}}(F_t \,\Vert\, U) \le D_{\mathrm{KL}}(F_t^K \,\Vert\, U),$$
where $U$ is uniform. To justify this, the authors invoke Cover & Thomas and write
$$H(F_t^K) = H(M_K(F_t)) \le H(F_t),$$
with $M_K$ the top-K masking operator, concluding that KL-to-uniform must increase.

This conclusion only holds if $M_K$ is a true probability-space top-K renormalization, where all probability mass outside the top-K is set to zero and the rest renormalized. In that case entropy decreases and KL-to-uniform increases.

In practice, however, the method applies masking in **logit space**: non-top-K logits are set to zero and softmax is taken over all classes. This does not eliminate the tail; each masked class contributes $e^{0/\tau}=1$ and receives a small but positive probability. With many masked classes, the distribution moves closer to uniform, entropy can **increase**, and the Cover & Thomas inequality no longer applies.

Thus, Proposition 1 is proved for a different operator than the one actually implemented, so the theoretical guarantee of "concentration" does not hold. Can the authors comment on this?

**2. Issues with Information Bottleneck (IB) analysis**
The paper writes IB objectives in a way that is technically incorrect, treating optimization as if it were over the variable $F$ itself rather than over the encoder distribution $p_\phi(F\mid X)$ that generates $F$. In the standard IB framework, the optimization is carried out with respect to the encoder parameters $\phi$, not directly over $F$. Moreover, in deterministic networks with continuous inputs, $I(X;F)$ is infinite unless noise or quantization is introduced. These issues are not acknowledged, however the results are presented considering the IB measurements as rigorous. Please explain this discrepancy.

**3. Mutual Information estimation**
The "information plane" plots are based on proxy estimators: a decoder is used to approximate $I(X;F)$ and an auxiliary classifier is used to approximate $I(Y;F)$. These are lower bounds whose values depend heavily on the capacity and training of the auxiliary heads. Without careful controls (e.g., architecture specification, held-out evaluation, retraining at each checkpoint, confidence bands), the plotted trajectories mainly reflect how well the auxiliary models train, not true MI. In addition, the values are not normalized, so axis scales are not interpretable across datasets.

**4. Equating dark knowledge with high $I(X; F)$**
The paper equates "dark knowledge" with high $I(X;F)$, but in the KD literature dark knowledge refers to structure in the teacher's output distribution. Likewise, the claim that maximizing $I(Y;F)$ is less important for KD is not substantiated, since student performance still depends on predictive alignment with labels. Statements about overconfidence and temperature scaling are also asserted without accompanying measurements such as entropy or calibration.

**5. Issues with experimental results presentation**
#### Fairness and clarity of experimental comparisons:
1. It is not always clear which SFKD components are active in each experiment. For example, Table 2 appears to use logit masking only for compared techniques, but "ours" could be using all 3 (logits, features and attention if relevant), but this is not stated explicitly.
2. Table 3’s “+Ours” may add extra supervision channels (e.g., feature or attention masking) that the baselines do not have. Without matched counterfactuals that use the same number of loss channels but no masking, it is difficult to tell whether gains come from masking itself or simply from additional supervision. This lack of clarity raises fairness concerns about the comparisons.

#### Mixed and borderline empirical gains:
While the experiment section shows some good results, many CIFAR-100 and ImageNet deltas are small and may fall within run-to-run noise. Without reporting mean ± sd, paired significance tests, or aggregate effect sizes, it is hard to judge consistency. Clearer disclosure of which components are active, matched-component baselines, and stronger baselines such as KD+Label Smoothing would make the empirical case more convincing.

**Requested Changes:**

Here are some of my proposed changes:
1. **Proposition 1 should be aligned with the actual operator.** Either probability-space top-K with renormalization (or $-\infty$ masking) should be adopted so the "concentration" proof applies, or the general guarantee shoudl be dropped and instead empirical measurements of entropy and KL-to-uniform before and after masking should be provided.
2. **IB analysis should be clarified.** The objectives should be recasted correctly (optimization over the encoder distribution $p_\phi(F\mid X)$), while acknowledging that $I(X;F)$ is ill-posed for deterministic nets, and presenting the IB perspective as a motivating lens rather than a rigorous derivation unless noise/quantization is explicitly introduced.
3. **Disclosure of MI estimation protocols.** Please specify the decoder/classifier architectures, training splits, convergence criteria, and whether heads are retrained at each checkpoint. Provide error bars or confidence bands. Without this, the "information plane" figures are not reliable evidence.
4. **Correct figure labeling.** In Fig. 2b–d, the y-axis labels are incorrect or mismatched with the quantities described in the text. These need to be fixed to avoid confusion.

**Recommended improvements (would strengthen the work):**

5. **Normalize MI axes and units.** MI values should be reported in interpretable units (bits/nats, per pixel, or as a fraction of entropy) so scales are comparable across datasets and architectures.

6. **"Dark knowledge", could be empirically evaluated** Dark knowledge is being equated with  $I(X;F)$ in a handwavy manner. Instead, the authors could use direct measures such as teacher–student confusion matrix similarity, calibration (ECE, NLL), or representation similarity. Or this line of claims could not be pursued as well.

7. **Define and justify K-selection protocol.** Clearly stating how K is chosen for each experiment (e.g., validation accuracy, entropy target) and providing ablations across K and temperature values would make the experiments stronger.

8. **Add masking ablations.** In the current framework I believe, probability-space top-K renormalization, logit zeroing, and $-\infty$ masking, reporting entropy, KL-to-uniform, calibration, and accuracy for each should be compared.

9. **Interpretive claims could be reframed** I believe the statements about "IB demonstrates overconfidence" or "maximizing $I(Y;F)$ is less crucial," should be toned down unless directly supported by experiments.

---

> ### Author Response · Authors · 2025-09-27
> **Response to Reviewer YK4L**
>
> ***Response to W. 1 \& R. Ch. 1:***
> Thank you for this sharp  observation. You are correct that our initial proof for Prop. 1 did not align with the logit-space masking operator in our implementation. While the proof was intended to provide intuition, we agree its technical inaccuracy is a argumentative flaw. Following your high-level advice to reposition the paper as a more robust empirical study, we have removed Proposition 1 from the manuscript. Instead of pursuing a formal theoretical guarantee that you rightly identified as feeling ``forced'', we now connect SFKD to the Information Bottleneck principle through our extensive empirical analysis and visualizations. We believe this approach makes the contribution clearer and more compelling by focusing on the method's practical simplicity and demonstrated effectiveness.
>
> ***W. 2 \& R. Ch. 2:***
> We appreciate you pointing out the technical inaccuracies in our presentation of the IB framework. Our goal was to use the IB principle as a motivating lens for our method, but we acknowledge our initial formulation was imprecise. We revised our discussion in Sec. 4:
> -  We have reframed the section to explicitly state that we use the IB framework as a guiding principle and a tool for qualitative analysis rather than a source of rigorous derivation.
> -  We have corrected the IB objective notation to clarify that the optimization is over the encoder parameters that generate the representation $F$.
> -  We now acknowledge that for deterministic networks, $I(X;F)$ is technically ill-posed and that our plotted values are practical estimates based on well-established proxy estimators.
>
> We believe these changes present the IB analysis in a more accurate and appropriately contextualized manner, aligning with your suggestions.
>
> ***W. 3 \& R. Ch. 3/4/5:***
> We appreciate the reviewer’s request for concrete architecture details of the MI estimators.
> Following established practice in Revisiting Locally Supervised Learning: An Alternative to End-to-End Training [InfoPro] (Wang et al., ICLR 2021),
> we adopt lightweight auxiliary heads to approximate $I(X;F)$ and $I(F;Y)$:
> -  ***Backbone.*** ResNet-34 trained on CIFAR-10 (the setting used for Fig. 2). Features are logged after each stage, and we report information-plane values from the deepest representation $F_5 \in \mathbb{R}^{512 \times 4 \times 4}$.
>
> - ***Decoder for $I(X;F)$.*** A compact upsampling--convolutional decoder, mirroring the InfoPro design:
>     $$
>     512 \xrightarrow{1 \times 1} 256 \xrightarrow{\uparrow \times 2} 128 \xrightarrow{\uparrow \times 2} 64 \xrightarrow{1 \times 1} 3,
>     $$
>     with skip-connections from intermediate features. The output is passed through a $\tanh$ activation and optimized with $\text{BCEWithLogitsLoss}$ against the unnormalized input image.
>     We report the normalized proxy $1-\text{AvgBCE}(x|F)$.
>
> -  ***Auxiliary classifier for $I(F;Y)$.*** A lightweight head consisting of two $3 \times 3$ conv+BN+ReLU layers, followed by global average pooling and a 2-layer MLP producing logits for 10 classes.
>
> This point is further addressed in our response to Reviewer CJQH’s R. Ch. \#4, where we provide additional details on how the mutual information estimator is updated during training. The labeling issue in the information-plane plots has been corrected.
>
> ***R. Ch. 7:***
> We addressed this in our response to Reviewer ZsE8’s R. Ch. 7, discussing performance sensitivity to different top-K values.
>
> ***W. 4 \& R. Ch. 6, 9:***
> We appreciate your feedback on our interpretation and terminology. You are correct that ``dark knowledge'' traditionally refers to the relational information in the teacher's output distribution, and our initial phrasing was imprecise. We have revised the manuscript to tone down and clarify these claims.
> -  We have reframed our argument to avoid equating ``dark knowledge'' directly with I(X;F). Instead, we now argue that indiscriminately transferring all information from the teacher (which has high $I(X;F_t)$ can introduce noise and redundancy, and that SFKD acts as a bottleneck to filter for the most salient, task-relevant information.
> -  We have removed  assertions about ``over-confidence'' and have reframed the discussion of $I(Y;F)$ to be more grounded in our empirical findings, as seen in the information plane analysis in Fig. 2.
>
> Our goal is to present a clean, empirically-supported argument, and your feedback has helped us remove speculative claims and focus on what our results directly demonstrate.
>
> ***W. 5:***
> We acknowledge the reviewer's concerns about experimental clarity and have addressed these issues in the revised manuscript: All instances of ''Ours'' have been replaced with specific method names (e.g., ``KD + SFKD'') to clearly indicate which SFKD components are active in each experiment. This issue is further discussed in our response to Reviewer ZsE8’s W. \#2, where we examine the statistical significance of our performance improvements.

---

### Decision · Action_Editor_4x3f · 2025-11-02

**Recommendation:** Reject

**Audience:**

Yes

**Audience Explanation:**

With some work, the question being investigated in this work is of interest to a sizeable audience in TMLR, as effective knowledge distillation is important.

**Claims And Evidence:**

No

**Claims Explanation:**

The reviewers raised several concerns, some of which were addressed by the updated paper. However, there remains issues around the significance of differences in the results and certain key details remain unclear.

The author responses included some description of low variability, but the manuscript itself does not justify if differences are significant and how they are determined to be significant. With a small number of runs per dataset, this might be difficult to do (e.g., with a paired t-test) but you could report significance across the datasets. It is currently over-claiming to state SFKD performs better, without a more rigorous justification.

A review also asked for an explanation for how K is chosen, but it does not seem like this was added to the paper. This should be explained in more depth, since K is such an important parameter for this algorithm. It would also be useful to also provide deeper insights into sensitivity to K.

For mutual information estimation, it is fine if this is a bit expensive, since it is only for understanding in the experiments (not part of the algorithm). The plots showing mutual information, though, were a bit hard to follow. It might be worth explaining in that section how mutual information is computed, on which variables, to discuss those plots in more depth. The small amount of description in the paper also does not include citations to how other work has done it, and better justification for why the choices made do not skew the results (as pointed out by a reviewer).

Finally, the reviewers reasonably pointed out issues with the theory, and it was removed. But the paper still says in certain places that certain connection to IB are proved.